# The Sustainability of Emerging Social Vulnerabilities: The Hikikomori Phenomenon in Southern Italy

**Vincenzo Esposito** [1,*] **, Felice Addeo** [2] **, Valentina D'Auria** [2] **and Francesca Romana Lenzi** [3]

1   Dipartimento "SARAS"—Storia, Antropologia, Religioni, Arte, Spettacolo, Università degli Studi di Roma "Sapienza", 00185 Rome, Italy
2   Dipartimento di Scienze Politiche e della Comunicazione, Università degli Studi di Salerno, 84084 Fisciano, Italy
3   Laboratory of Psychology and Social Processes in Sport, University of Rome "Foro Italico", 00135 Rome, Italy
*   Correspondence: vi.esposito@uniroma1.it

**Abstract:** We can classify the Hikikomori phenomenon with the classification of "social pathology": the Hikikomori phenomenon, and its spread in society, appear to be a real danger to the sustainability and resilience of the very society in which it occurs. This is because the social isolation of an individual, especially if young and non-independent, impacts the community of reference in human, economic and psychological terms. Therefore, an analysis that investigates the social aspects of the Hikikomori phenomenon cannot disregard the fact that it can be said to be sustainable in the community of reference within which it occurs. This, without wishing to produce a judgment on the merits of the social pathology, is relevant to assessing the capacity of that community to sustain its presence and spread and the human and social costs required to contain it. The research aims to explore the Hikikomori phenomenon in the context of Southern Italy, considering it as an emerging social vulnerability that impacts very deeply onto the sustainability of a social, economic and community systems such as the Campania region one. The following paper therefore presents empirical work conducted in southern Italy, in the Campania region. The methodology used is Mixed Methods, and the research design is Sequential Exploratory. The respondents were reached through the help of the association Hikikomori Italia.

**Keywords:** Hikikomori; mixed method; focus group; Delphi method; sustainability

## 1. Introduction

Hikikomori is a Japanese term formed from two words "Hiku" (to pull) and "Komuru" (to withdraw), which means "to stand aside, to withdraw" [1,2]. This term is first used with scientific value, to refer to both the syndrome and the sufferer, by psychiatrist Tamaki Saito in 1998 in his work "Endless Adolescence". In his book, the author takes a radical stance by pointing to Hikikomori as a new form of isolation and a new psychopathology that cannot be superimposed on any other known ones.

In 2003, the Japanese government, through the Minister of Health, designated Hikikomori syndrome as new psychopathology and identified its characteristics: (1) continued isolation for 6 or more months; (2) motivations for isolation, dictated by the desire to escape the pressures of social fulfillment; and (3) distrust of relationships, the individual who decides to isolate himself develops a strong distrust of interpersonal relationships, society and its dynamics.

In the first research (medical in nature) that also consisted of treatment, Hikikomori was confused with depression or schizophrenia and treated with psychotropic drugs: most often with poor and inconclusive results [2–4]. After identifying Hikikomori as a new psychopathology and creating ad hoc therapies for its treatment [1,4], today there is a tendency

to confuse this syndrome with two other disorders, Internet Addiction and Gaming Disorder. While in these cases the isolation is due to an addiction to the Internet or video games, in the case of a Hikikomori the isolation is due to social malaise [2,5]. According to Ricci and Pierdominici, only 30 percent of individuals who self-isolate are offline and it is estimated that only one-tenth of the total number of individuals make heavy use of the Internet [6,7].

Hikikomori syndrome has been classified in the 2019 version of the DSM as a cultural syndrome: this type of label is given only when a certain pathology affects only a state or a single population. In the 2022 updated version of the DSM, Hikikomori syndrome is included in the appendix of psychopathologies ready to become an integral and fixed part of the volume.

The work presented here, therefore, is a survey conducted at the beginning of the year 2020 in the Campania region (South Italy) that aimed to investigate the situation of the Hikikomori phenomenon in this part of Italy and whether it was sustainable, or unsustainable for the families and community of reference, the presence of this phenomenon in an area such as Southern Italy.

The impact of globalization is essential for the proliferation of the Hikikomori phenomenon which in turn exerts a considerable impact on the sustainability of the welfare network, such as the social assistance. This is a research field that includes impacting studies, as the ones that apply Community Engagement and Vulnerability Assessment's tool to study social determinants of health [8]. To talk about sustainability means to convey a definition of world, society and community that improves the generalized well-being, the appropriate and sustainable allocation of resources, which includes a world that does not abandon individuals and their network of proximity (families, communities) to the margins. The "ecology" of a system—therefore, the sustainability of its functioning—does not only concern the environmental dimension, but also the social and human one. This declares a community's progress level and determines its future scenarios [9]. This refers to developments in cultural diversity, tradition, social systems, globalization, immigration and settlement, and their impact on cultural or social sustainability. The Hikikomori phenomenon, like many others, is an example of cultural transition in the evaluation of social marginality. One of the results of the processes of evaluation and study is influencing them as a decisive issue in social policies, rather than ignorance and confinement in a gray area and social invisibility. This consists precisely in the determination of parameters of social sustainability of a country's system [10].

## 2. Literature Review

In the 1970s and 1980s in Japan, for the first time, several studies identified symptoms of "withdrawal neurosis" [2,11] in situations where patients decided to withdraw from their everyday contexts, such as school or work. Early cases of withdrawal (occurring mostly among young people) were often confused with various existing psychopathologies, such as schizophrenia or depression. Due to the increasing attention paid to these symptoms by various scholars, more in-depth studies of the disorder began to be conducted, eventually recognizing it as a real syndrome different from all others already known [5,11,12].

In the 1990s, these symptoms were traced to the syndrome that is now known and identified as Hikikomori [2,3,5,13,14]. Since voluntary isolation is also a prodromal symptom of schizophrenia, in early diagnoses, Hikikomori syndrome was treated with the drug and nursing therapies provided for schizophrenic patients [2,5]. However, in Hikikomori, all of the symptomatology that greatly alters the perception of reality present in schizophrenics is absent. Far from being considered persons incapable of understanding, those who become Hikikomori first and foremost choose it; that is, they decide it with lucidity [15,16].

According to some authors [11,13,16,17], Hikikomori can be classified as a true state of anomie, a term coined by Durkheim [18] that means "absence of norms". However, their loss of normative references is not related to the general context, but only to those aspects related to

social dynamics, personal success and one's public image. It would thus be a form of narrow anomie, as other norms, such as moral ones, are maintained.

One of the first general characteristics that emerge from early studies on Hikikomori syndrome is the slow course of the pathology, which does not manifest itself suddenly and blindly, but as a long process. Crepaldi [19] tried to identify the stages that a Hikikomori goes through before total isolation. What can be inferred is that the isolation resulting from Hikikomori syndrome is not a choice dictated by a bad personal moment. It is a reasoned choice that the person matures over time and in full clarity. In the first stage, the person begins to feel the urge to isolate himself or herself, finds relief in isolation and feels uncomfortable with other people; in the middle stage, the person begins to consciously process the impulse to isolate, quits school or work, reverses the sleep-wake cycle [3,20] and begins to spend most of their time in their own home [19].

Among the main reasons why young people become Hikikomori is the pressure for social achievement and the race for perfectionism [2,5]. In young people, the onset of Hikikomori disorder often coincides with the end of school age (age 19) and the beginning of adulthood, when important decisions about one's future must be made. At this delicate stage, those who have been termed "emerging adults" [16,21], ages 18–29, must build their future and identity. In high-income societies, young people are often under constant stress caused by the competitive climate at both school and work. In this regard, other reasons related to the sociocultural context of young people in self-isolation have been highlighted: frequent incidents of bullying suffered and bad relationships with peers [13,22]; the pressure exerted by schools and inattention of educational institutions toward young people who are Hikikomori or prone to become Hikikomori [4,11,23]; and an overly protective and comforting family context [2,4,5,23].

Japan is considered the homeland of the Hikikomori phenomenon [24,25], and to date it remains the country with the largest number of isolated children. Data dating back to 2011, reported in a study by Takasu et al. [20], showed that about 200,000 people in Japan can be classified as Hikikomori (out of a population of about 120 million). One of the reasons that could be attributed to the phenomenon is certain Japanese cultural traits. From a cultural point of view, in Japan great importance is placed on "sekentei" (世間体), which means "appearance in the eyes of others" and reflects social expectations that the individual cannot disappoint. Therefore, dependence and conformity in Japanese culture are part of cultural values that can guide and condition each individual's being in society. According to several authors [4,6,16,23,24,26], the boys who are most likely to become Hikikomori are those who prefer to spend a lot of time alone, have an economically well-off family, perform well in school, and are often bullied. Along with these "personal" factors, there are some contextual factors that facilitate withdrawal; school and/or university pressure, fear of facing new realities (many Hikikomori decide to isolate themselves at the end of a school career, ready to start a career path), and finally Japanese work environment, which is very competitive. As for other nations, we do not have many studies exploring these characteristics.

In 2014, Fansten et al. [27] hypothesized the existence of different blocking modes in the socialization process of Hikikomori. Their study proposed the construction of a typology to define the different types of Hikikomori, later revised by Crepaldi [19]. Currently, from the typological classifications developed by these scholars, four types of Hikikomori can be distinguished:

1.  Alternative: Those who belong to this type decide to isolate themselves because they do not accept the social dynamics of modern society. Individuals categorized in this way isolate themselves to avoid "regular" adolescence and rebel against society. Is possible that in this case, the boy before reclusion is preceded by a severe existential depression;

2.  Reactionary: Those in this group experience an many times familiar, context, already difficult one that has contributed to exacerbating an already pre-existing. Imprisonment is a reaction to their already difficult situation and is linked to an especially traumatic events;

3. Resigning: This type of withdrawal concerns those individuals who cannot withstand the pressure arising from others' expectations. The subjects manage to alleviate their malaise by isolating themselves by abandoning social competition, whether in school and work settings;

4. Chrysalis: The person in this typology seeks an escape from the responsibilities of future adult life in isolation. He feels that he does not have the skills to deal with the future, which is seen as a source of anxiety; therefore, every thought about it is avoided through the mechanism of avoidance. The Hikikomori is like freezing time and adopting strategies, unconsciously or consciously, aimed at flattening one's life and trying to freeze the present. Here, we will have the reversal of the rhythm of sleep-wake and the preventing of light from entering the place used for isolation. There is an alteration in the mechanisms of nourishment, eating meals in a way that is fast and irregularly.

On the other hand, Li and Wong [26] proposed a classification of the Hikikomori syndrome about whether or not the person is affected by other diseases. In this regard, the distinction between primary Hikikomori and secondary Hikikomori has been proposed. In the former case, the syndrome is not related to any pre-existing psychopathology of the person. Secondary Hikikomori, on the other hand, is when the confinement is a direct consequence of an already present problem, such as anxiety disorders or obsessive-compulsive disorders.

Over the past decade, studies on Hikikomori syndrome have spread outside Japan and around the world, allowing for a deeper understanding of a condition that is as silent as it is extensive. Major studies include those conducted in France [27]; in Italy [19]; Canada [28]; Australia and South Korea [29]; Spain [30]; Belgium [31];

Although studies have spread to most states in industrialized society, there is still a gap that has not been closed. Although Hikikomori disorder was initially treated with depression and anxiety, this is fortunately no longer the case today. However, there is still a lack of clear diagnosis to identify ad hoc treatment for this type of patient [13].

Some recent studies have pointed out the strong relationship between Hikikomori and the use of technological devices, especially the Internet. The development and spread of the Internet have influenced the behavior of people living ordinary lives and that of Hikikomori. While early case studies showed no link between Hikikomori and technology use habits [5], more recent studies [26,29,32,33] show the opposite: Hikikomori spend most of their day online, mostly playing online games, developing in many cases what is now recognized as Internet Gaming Disorder (IGD) and Internet Addiction (IA). The relationship between these disorders and the pathology of Hikikomori continues to intensify, as evidenced by studies such as that of Tateno et al. [34], which show an increase in the amount of time spent online. In recent years, the intensified coexistence of these disorders has led to confusion, resulting in the overlap between them, to the point that the pathology of Hikikomori is confused with both Internet Gaming Disorder and Internet Addiction, and conversely [29,33,35]. For Hikikomori, the Internet is an escape route since using it they can create new relationships and use that Web-mediated sociality as a substitute for the sociality lost by isolating themselves at home [34,36,37].

## 3. Research Question

Some research in the literature [38] shows an increasing number of Hikikomori, especially in wealthy and developed settings. These types of environments push young people into the constant race to get a good degree, a good job, and a family. This would push the most vulnerable individuals to isolate themselves and give up the perennial competition with peers, unable to withstand the ever-increasing social pressure. From an epistemological and empirical point of view, this kind of perspective leads to more and more studies being conducted in cities of a certain type (wealthier and more emancipated) rather than in others. For example, in a country such as Italy, where the gap between northern and southern regions is still alive, the latter is perceived as a more backward context, thus theoretically less prone to the development of the Hikikomori syndrome. This study aims to investigate the Hikikomori

phenomenon precisely in a region of southern Italy (Campania), where there is a great lack of studies on the subject.

This study aims to investigate the presence of the Hikikomori phenomenon in Southern Italy and whether it is sustainable for the family structure and community where it occurs. To achieve this goal, the chosen approach is that of a descriptive-exploratory study. However, given the impossibility of personally involving those affected by the Hikikomori syndrome, the experiences of various social actors closest to them were cross-referenced. In particular, parents and experts were identified and involved, as they were considered useful sources to detect information that unfortunately was not directly detectable. In the case of parents, information was collected on the parental experience of families with Hikikomori through a focus group aiming to study everyday family life and trace any differences and similarities in Hikikomori behavior. In the second case, professional figures such as psychologists, behavioral technicians, and child neuropsychiatrists were involved through the Delphi technique. The involvement of experts in our empirical study was essential to identify the characteristics common to Italian Hikikomori and to construct predictive scenarios that could help the same specialists recognize and deal with the syndrome.

## 4. Methodology and Materials

Given the complexity of the phenomenon and the heterogeneity of the actors involved, a methodological approach capable of integrating different empirical perspectives within the same study was used. The methodology used was Mixed Methods [39].

First used in 1959 by Campbell & Fiske to study the reliability of certain indicators that were used to assess the mental health of certain psychological patients, they have emerged over the past 50 years as a viable alternative to the Qualitative (henceforth QUAL)—Quantitative (henceforth QUAN) dualism. Campbell and Fiske's is not the first research in which more than one method has been used at the same time, suffice it to say that in 1933 Lazarsfeld and Jahoda conducted research on the unemployed in the village of Marienthal in Austria using 17 different data collection methods. In this case, as in other research, however, we do not speak of Mixed Methods (henceforth MM) but of the Multi-Method because the two Austrian authors after collecting the information did not use data integration techniques unlike the work conducted by Campbell and Fiske, who were the first to integrate the different data collected [39,40].

In the classical perspective of use, MMs are an ideal combination of between QUAL—QUAN approaches. Over the years, research designs have been developed that included two or more QUAL or QUAN methods [41].

Mixed Methods' special feature is constructing compound and articulated search designs. In this case, we speak of two types of research designs: simultaneous, in which the various steps of research are conducted simultaneously, and sequential, in which the results of the first step are used to construct the second step and so on [39,42].

Four basic research designs are used within MM: Parallel Convergent; Nested; Exploratory Sequential; and Explanatory Sequential. In this project of ours, the research design used is the one called Exploratory Sequential where the first phase of the research is given a higher priority and then concludes with a different research method constructed using data from the previous phase [39,42].

Initially, the research was planned with three phases, an initial part was formed by the focus groups. After conducting this phase and analyzing the results, we used the results of this phase to create the outline of the first phase of the Delphi Survey. The last planned phase was the phase formed by the questionnaires that would be created using the results from the Delphi Survey. Due to the tight time frame we had to finish the work presented here and the arrival of the pandemic, the last phase could not be conducted, so the first two phases were conducted. Therefore, in this case we will be dealing with a Delphi called confirmatory [43], where a maximum of 2 steps are conducted and whose goal is to confirm the results of the focus group through expert opinion.

Therefore, the methods integrated within the research design are the focus group and the Delphi Survey.

Focus groups are a group interaction where what is crucial is not only the answers of the respondents but the interaction that arises around the answers given by the respondents [44]. Originating in 1941 through research by Lazarsfeld and Merton over the years, they have emerged as a very fruitful method of information gathering used especially in market and marketing research. This technique will be used because it offers several potentials: it is particularly suitable for detecting unexpected aspects of a phenomenon since it highlights the reference patterns of the subjects studied [45,46]; it is suitable for obtaining a lot of information and a sufficient level of depth in a short time and at low cost and is therefore very useful for studying an unknown phenomenon [45];and some argue [46] that interaction among participants can foster a change of opinion and the formation of ideas to those who have none on the subject. The basic elements of this method are the group of participants, from which the interaction will be created; the moderator (or conductor) of the focus, who manages the interaction and determines its success; and the observer, who assists the moderator while conducting the focus group [46,47]. In this case, we used focus groups with the parents of Hikikomori children that we recruited thanks to the Hikikomori Italia association.

The protocol interview was constructed after a careful review of the literature to identify the most important dimensions studied in the literature. Based on Teo's (2018) work, we studied the dimensions considered by the authors who constructed the questionnaire and created open-ended questions that allowed us to study the everyday life of Hikikomori families in southern Italy. The constructed protocol explored three main dimensions aimed at investigating the families' daily life, tracing the biographical path of the isolated boys, and investigating how belonging to a self-help group improved their daily life.

The second part of the research was conducted with a Delphi survey. Generally, the Delphi survey is not used to collect the views and opinions of experts, but to build scenarios on issues and problems for which there are no databases or trends, and therefore specialists, really knowledgeable people, are the only ones who can correctly answer and satisfy our purposes [48]. Delphi inquiry is presented as an iterative method that unravels in several stages. Usually, there are three stages, but it is not uncommon for it to require a fourth [48,49]. The fundamental prerogative of this technique is that the different experts who are part of it never come into contact with each other and ignore who it was that gave certain answers. This strategy is intended to prevent participants from constructing a judgment based on any presumed academic, political, or scientific antagonism that might lodge among them [43]. This type of technique offers many potentials such as being able to create plausible future scenarios and arrive at an understanding of a phenomenon by exploiting the theoretical backgrounds and field experience of experts. However, this technique is not without its limitations. The most obvious limitation is that being a set of questionnaires it is not unusual for some experts to stop responding in advance of the end of the technique [43]. In this case, we exploited avalanche sampling procedures. This procedure consists of randomly selecting different units of the population we are interested in questioning who are asked to indicate other units that belong to the population of interest [41].

In this case, the Delphi protocol was constructed from the analysis of the previous phase. After analyzing the results of the focus group, we identified some topics that needed to be explored in depth with the help of the experts. The Delphi protocol had the following as main topics: the search for a definition shared by experts regarding the Hikikomori syndrome; the causes of the syndrome; and the role of school in the development of the syndrome.

During the analysis of the results, excerpts taken from the focus group interview or the Delphi Survey will be presented. The sentences, we have decided to present, will be included in the text either in the original language, Italian, and translated into English. This choice was made to avoid as much as possible the loss of information that results from translating a passage from one language to another [50].

## 5. Results

The results of the empirical work in this study will be divided into two sections, one devoted to the focus group and one to the Delphi technique.

*5.1. Focus Group*

The interviewees were collected with the help of the Hikikomori Italia Association, an association that has been active on the Italian peninsula for several years.

The first part of the Focus Group served to investigate the lived experience of families with a Hikikomori person and their behavior once isolated. The respondent group consisted of 12 people, of whom 8 were women and 4 were men. The interviewees confirmed, in agreement with the literature [2,24], that the process of isolation is a gradual process that the parents themselves often do not notice:

> *"Non ci si rende mai conto dell'inizio, perché è così graduale, comincia in un modo…piano piano piano piano, questi ritiri sono dei piccoli passi indietro non ci si fa neanche caso".* (Donna, 42 anni)

> *"You never realize the beginning because it's so gradual, it starts one way...slowly slowly, these retreats are small steps back you don't even notice."* (Female, 42)

This gradual onset is often accompanied by the child's disaffection with human relationships and the choice not to continue attending school, considered by the latter to be an anxiety-provoking factor.

The consequence of this decision to isolate themselves leads to a disruption of family habits. This change within their daily routine leads parents to a more hurried approach to everyday tasks. The primary thought is to return home to check on their child. This situation leads to "abandonment" of what should be normal for a family with a teenage child:

> *"Non c'è più la quotidianità di prima, non c'è l'andare a parlare con un professore, non scambiarsi il numero di telefono con un'altra mamma".* (Donna, 45 anni)

> *"There's no more daily routine than before, no going to talk to a professor, no exchanging phone numbers with another mom."* (Female, 45)

This change is reflected not only in the human relationships of the boy who decided to isolate himself, but also in the work and social relationships of the parents. Once the boy has isolated himself, it becomes untenable for them to relate to relatives and colleagues since their only thought is to return home to take care of their son.

Human relationships also change. In this case, one has to deal with the estrangement of friends, but many also complain of family members' lack of interest in what is happening to their son:

> *"la famiglia sparisce…sparisce perché?...Parenti e amici ti chiedono, non hanno delle risposte e non comprendono e si allontanano e non vengono a bussare…non vengono più a bussare alla tua porta, anche le mie sorelle quindi le zie a dire "come stai?" "che fai?"* (Uomo, 50 anni)

> *"family disappears...disappears why?...Relatives and friends ask you, they don't have answers and they don't understand and they move away and they don't come to knock...they don't come to your door anymore, even my sisters so aunts to say "how are you?" "what are you doing?"* (Male, 50)

According to our interviewees, there is a difference between the decision to isolate oneself at home or the decision to remain cooped up in one's room. In the second case, boys refuse to leave their "den". Very complicated was the collection of information regarding the boys' daily routine because very often parents are in the unawareness of what their children do during the day. According to them, however, the Internet is central when experiencing this situation because for many it is the only way to form relationships with people outside their home. The moment the period of isolation begins and one is abandoned by one's friendships, the web is a place to form bonds. Although they are not aware of the specific activities, the certainty is

that most of them are through the use of the Internet (playing online video games, watching TV series, etc.). For many of these children, the phone is the favorite companion during the day. Moreover, according to the interviewees, these activities are conducted by the children during the night hours because the isolation is often accompanied by the altered sleep/wake cycle.

In the second part of the focus, interviewees talked about their children's journey. The focus was mainly on children's relationship with school. The latter is considered by parents as the "*society*" their children have to deal with, the first approach outside the home walls and away from the parental nest.

School performance is not a problem for Hikikomori: grades turn out to be good, often better than their classmates. However, for some, withdrawal coincides with the approach of the final exam or a move to another institution. This may be a testament to a willingness not to change their condition, by an immobility in which well or poorly they manage to stay afloat. The real problem for them is the relationship with peers within school structure.

The presenters talked a lot about the phenomenon of bullying, both from the boys' classmates and from teachers. This type of bullying is more difficult for those who suffer it to digest because it comes from those who are supposed to protect you within the school environment. Yet, all those present said they have encountered people who are insensitive and unwilling to engage in dialogue with their children:

> *"una delle ultime volte che ha avuto un problema con una professoressa lei mi disse questa frase: mamma se questa cosa fosse successa con uno dei miei compagni io avrei saputo come reagire, perché ci sono abituata. Ma di fronte alla professoressa che potevo fare"* (Donna, 52 anni)

> *"one of the last times he had a problem with a teacher she said this sentence to me: mom if this thing had happened with one of my classmates I would have known how to react, because I am used to it. But in front of the teacher what could I do"* (Female, 52)

According to many of the parents, these behaviors are repeated and systematic. The boy is targeted, and this leads to an inability to react given the difference in status between the two individuals.

It emerged from the discussion that the individuals mentioned had in school the only contact with "outside society" (where outside means the world they are about to experience outside the quiet family), extracurricular activities were few and outings with friends very rare.

Interviewees were asked whether they knew the exact moment when their son decided to isolate himself, and thus whether they knew of any particular episodes that represented the so-called breaking point. For the interviewees, there is no one key moment, but many small moments that lead to the break with society.

The scenario that emerges is that parents feel guilty about this situation, especially for not realizing in time that something was wrong:

> *"me lo sono chiesta moltissime volte…mi sono chiesta se la separazione con mio marito avesse influito, se l'avesse in un certo senso reso ancora più fragile…le domande sono mille, ogni genitore si chiede "dove ho sbagliato?", "che cosa ho fatto di sbagliato?"…me lo chiedo tutt'ora…tutt'ora mi chiedo se ho sbagliato qualcosa, se continuo a sbagliare…non lo so…cioè ritengo di aver fatto bene delle cose, cioè tante cose, ovviamente come tutti possiamo commettere errori, forse non l'ho capito…forse stava vivendo il disagio e non me ne sono accorta perché magari presa da tante situazioni, il lavoro, la separazione, tantissime cose, quindi magari non mi sono accorta di lui…non lo so"* (Donna, 52 anni)

> *"I have asked myself this many times...I wondered if the separation with my husband had affected, if it had in a way made him even more fragile...the questions are a thousand, every parent wonders "where did I go wrong?", "what did I do wrong? "...I ask myself to this day...to this day I wonder if I did something wrong, if I continue to do wrong...I don't know...that is, I think I did some things right, that is, so many things, of course like everyone we can make mistakes, maybe I didn't realize it...maybe he was experiencing discomfort and*

*I didn't notice because maybe caught up in so many situations, work, separation, so many things, so maybe I didn't notice him...I don't know"* (Female, 52)

Finally, other interesting themes that came out through the interaction due to the focus groups are the issue of how boys feel in awe of their bodies:

*"continua a dire "io non ho più gli amici perché, all'epoca non ero bello e quindi non ero accettato, e quindi per questo adesso non me la sento" …continua a dire non me la sento, se mi cerco adesso degli amici faccio la figura dello sfigato."* (Uomo, 56 anni)

*"he keeps saying "I don't have friends anymore because, at that time I wasn't handsome and so I wasn't accepted, and so that's why I don't feel like it now" …he keeps saying I don't feel like it, if I look for friends now I look like a loser."* (Male, 56)

For our interviewees, the phenomenon of Hikikomori has become central to their lives, not only because they have a loved one living with this condition, but also because of their affiliation with the Hikikomori Italia association, thanks to which they have found people to talk to and who do not judge:

*"poi è nata l'associazione è per quanto mi riguarda siamo rinati tutti, piano piano"* (Donna, 48 anni)

*"then the association was born and as far as I'm concerned we were all reborn, slowly"* (Female, 48)

Finally, according to the testimonies collected, the parent-child relationship improved when the former stopped commenting on the latter's way of life:

*"diciamo che abbiamo capito che non giudicarli, che loro fuori si sentono giudicati…e noi lo facevamo involontariamente perché lo subivamo il giudizio degli altri e quindi giudicavamo anche noi all'interno della casa lui…quindi il non giudicarlo più per come vive la sua vita l'ha portato a confidarsi ad aprirsi, almeno verso di noi, questo che prima non succedeva"* (Donna, 41 anni)

*"let's say we understood that not judging them, that they outside feel judged...and we were doing it unintentionally because we were suffering the judgment of others and so we were also judging him inside the house...so not judging him anymore for how he lives his life led him to confide in opening up, at least to us, this that didn't happen before"* (Woman, 41)

Potential Limitations of the Approach Used

One possible limitation of this approach is the composition of the sample; the results reported above are the result of an interaction that developed from 10 people chosen based on criteria and characteristics considered useful for the research. Therefore, we used a reasoned choice sampling technique [51]. However, the results may not necessarily be the same with a group composed of different people in a different Italian region or a foreign country. Therefore, these results are difficult to generalize, but they reconstruct a fairly uniform scenario regarding Hikikomori present in Southern Italy.

*5.2. Delphi Method*

The Delphi panel was constructed using avalanche sampling [50]. We asked our contacts within the Hikikomori Italia association to indicate other experts in the field willing to answer our questions. Using this procedure, we interviewed 16 experts on the Hikikomori phenomenon. The questionnaire consisted of 4 obligatory questions and 1 optional question. Finally, there was a box where you could enter your first and last name or your email so that you could be contacted for the next steps.

The question with which the Delphi Method questionnaire opens was aimed at finding out whether there is a shared definition among experts in the field regarding the phenomenon of Hikikomori and to indicate what they consider to be some of the most relevant traits that distinguish this phenomenon.

In addition to withdrawing from social life, and locking oneself in the house, another trait is to eliminate contact with the outside world. This is the opinion of all respondents, indicating that the ties that are lost are not only those outside the family, but also those formed within the home:

*"Sottrarsi alle interazioni e relazioni dirette familiari e extrafamiliari, dedicarsi a attività totalizzanti da svolgere principalmente tra le mura domestiche o addirittura nella propria stanza"* (Psicologo, 45 anni)

*"Subtracting oneself from direct familial and extra-familial interactions and relationships, devoting oneself to totalizing activities to be carried out mainly within the home or even in one's room"* (Psychologist, 45)

Some experts also sought to identify common characteristics of these adolescents who choose to self-isolate. Those who answered along these lines identified frailties due to probable narcissistic wounding of their identity, low self-esteem, and strong social pressure. Finally, some argue that new technologies, the Internet, and video games somehow go to reinforce this impulse toward isolation.

The next question aimed to detect whether, according to the experts, there were differences between the Italian Hikikomori and the standard Japanese Hikikomori. The majority of experts identify differences due to the environment in which the phenomenon developed. This is because first of all, there is no unambiguous way of codifying the syndrome that is accepted by the countries in which it occurs:

*"Ne estistono, altresì, per l'assenza di un protocollo unico di gestione e di strutture adeguate, di strumenti pubblici e disponibili di raccoglimento delle istanze dei familiari. In definitiva, ne esistono a partire dall'assenza di una codifica universalmente condivisa della sindrome, qualora essa fosse possibile."* (Psicologo, 32 anni)

*"They exist, as well, because of the absence of a single management protocol and appropriate structures, public and available tools for collecting family members' petitions. Ultimately, they exist from the absence of a universally shared codification of the syndrome, should it be possible."* (Psychologist, 32)

Some identify Italian factors such as distrust in the future of the Italian context and institutions. Added to this are the country's economic situation and the typical Italian family model, all of which can accentuate marginalization within Italian society and thus lead to isolation:

*"Questo fenomeno potrebbe presentarsi laddove si incontrino profili psicologici con fragilità emotive, con bassa autostima e autoefficacia"* (Psicologo, 38 anni)

*"This phenomenon could occur where psychological profiles with emotional fragility, low self-esteem and self-efficacy are encountered"* (Psychologist, 38)

These personal emotional frailties clash with the pressures exerted especially in the school and work environment.

Regarding the family context, some identify similarities between the families of these boys, with professional parents, a very high sociocultural level, and with a very pressing presence of the mother, leaving the father with a marginal role within the household, in line with what the literature states [2,4].

The last mandatory question's goal was to identify what experts believe may be possible strategies to help and overcome the condition of Hikikomori.

A starting point to begin creating ad hoc strategies to avoid the isolation of these children is undoubtedly the coding of the syndrome. Having clear defining aspects it is possible to develop appropriate psycho-educational strategies:

*"Anzitutto la definizione della sindrome, la codificazione di essa, consentirebbe il riconoscimento e lo sviluppo di successive strategie condivise usufruibili."* (Psicologo, 40 anni)

*"First of all, the definition of the syndrome, the codification of it, would allow the recognition and development of subsequent usable shared strategies."* (Psychologist, 40)

The support strategies that experts propose are first and foremost a gradual reintegration into socialization. The most important thing is not to bring him out of isolation immediately. Just as the process of isolation turns out to be gradual [1,2,24], the opposite process must also be according to experts:

> *"L'obiettivo iniziale non è quello di far uscire il ragazzo di casa, quanto piuttosto iniziare ad avere ogni giorno un piccolo contatto con lui, entrare nel suo mondo, nei suoi interessi carichi di significati. L'accompagnamento verso un reinserimento sociale, scolastico o lavorativo avverrà gradualmente attraverso l'esposizione graduale"* (Psicologo, 39 anni)

> *"The initial objective is not to get the boy out of the house, but rather to begin to have a little contact with him every day, to enter his world, his meaning-laden interests. Accompaniment toward social, school or work reintegration will occur gradually through gradual exposure"* (Psychologist, 39)

Experts also suggest support measures for the family; some suggest a mentor with multi-disciplinary experience for the family along with psychological support interventions to help the household cope with this situation.

The last question was not obligatory and aimed to find out personal views on Hikikomori syndrome. Due to the optional nature of the question, only one response was received identifying Hikikomori and Internet addiction as two new pedagogical challenges for ages 10–18.

Potential Limitations of the Approach Used

In this case, the limitations and problems encountered mainly concern the sample used and the responses received.

Regarding the panel of experts to whom we sent the questionnaire, the main critical issue is that not all of them responded. In a Delphi survey, this is quite common. Bezzi [2] tells us that typically in the first round of this survey there are up to 20 percent defections. In this case, out of 16 questionnaires sent out, those received completed only 10, so the non-response rate is close to 37 percent.

Regarding the responses received in this case, the problem that arose was one of content. Some of the responses received were insufficient in terms of the argumentation required by the question. In some cases, the answer consisted only of a sentence without explaining why that opinion was held. For example, in the answers received for question 2, some were difficult to interpret because of the low information load, but one in particular was practically useless for the analysis of the results because the answer given was simply "no".

Finally, of the three planned rounds due to timing problems and the onset of the pandemic, only one out of three rounds was conducted.

## 6. Discussion

The current scientific literature on Hikikomori, and largely developed in Japan, agrees that Hikikomori develops in the presence of these traits: introverted person, hostile school environment, suffered bullying, and relationship with technology.

Concerning the prominent characteristic of Hikikomori, in both techniques, the samples spoke of shy, introverted people with low self-esteem, very similar to the Japanese cases [16,23,52].

Some studies [6,16,36] point to shame as the main feeling that drives Hikikomori to shut themselves in. However, what was recorded in the focus group, considering that parents' knowledge about the real reasons that drive their sons and daughters to isolate themselves is not exhaustive, is that Italian Hikikomori isolates themselves out of fear of being judged. According to experts and parents, a large proportion of Hikikomori are unable to accept their bodies. This is another breaking point with classic Hikikomori because Japanese studies do not seem to focus much on this point.

Among the different themes, results of different stages showed that school plays a crucial role. In agreement with other work already found in the literature [4,11,37], both experts

and parents identified the school as one of the main drivers that leads young people to develop Hikikomori. Another problem related to the school environment that emerged is bullying [16,36]. Those who decide to isolate themselves usually have relationship problems with peers, as has been found in several studies [15,16,23]. A very innovative output, however, is the description of some teachers as very similar figures to bullies for adolescents. In various papers [53,54], it is detected and studied how the teacher plays a crucial role in the growth and development of their students. Otherwise, from what emerges from our study, on the other hand, teachers are perceived sometimes as bullies. They are heedless of their students' frailties and often prone to criticize and worsen negative situations with their mistakes. In this regard, what parents are asking is that in the school context the relationship with Hikikomori boys and girls be handled differently without the latter being abandoned. The school system could routinely provide individualized curricula so that Hikikomori can continue their studies more easily, but this rarely happens.

The Hikikomori phenomenon turns out to be unsustainable for the family structure and community in Southern Italy. Many parents complained about the problem of having to give up meeting friends and relatives for a variety of reasons. In addition, after the workday, parents would return home to assist their children or try to assist them. Everyday family life was a hotly debated topic and was hotly debated by the interviewees. Some of the parents who participated in the focus groups identified shortcomings in their role as parents, often due to a lack of understanding of their child's problems and an underestimation of the situation. Experts interviewed through the Delphi method also seemed to agree with this, which is why many of the necessary interventions they reported were based on family support or parent training.

During the focus group, one of the topics that was discussed that was not included in the interview protocol was the point at which the boys decided to become Hikikomori [12,16,23]. According to the parents interviewed many of the boys had a critical point of no return where they decided to isolate themselves. Isolation, according to parents, always begins with dropping out of school. However, many respondents agree that is only the final part of dropping out of all the social relationships their child had. Although parents are aware that there was a specific moment and a specific episode that fostered social withdrawal, they do not know the specific episode because their children did not share this with them, pointing out that communication is not the best between the two parties. This is in line with previously published studies [2,5,16,36].

The relationship with technology is the last prominent trait that our research allowed us to detect, in agreement with the literature. Although "classic" Hikikomori, meaning those from the period between the 1980s and 1990s, did not use technology and particularly the Internet (since the latter did not emerge and develop until later), our sample stated that young Hikikomori make intensive use of technology and especially the Internet. In line with other recent studies [26,29,32,37], according to both focus group respondents and the Delphi survey, Hikikomori spend about 12 h a day on the computer, often using fictitious avatars to socialize with other people. According to experts, this practice could help young people unlock a new form of socializing that is alternative and complementary to offline socializing. In early research on the Hikikomori syndrome, the relationship with technology was underestimated or even denied, this was probably because there was not such widespread use of the internet as there is today. Today's studies [52,55–58] focus a lot on the relationship with technology, especially the Internet as it is a way for Hikikomori to create surrogate sociality to replace classical sociality. Our research is in line with this line of research.

## 7. Conclusions

A full understanding of the Hikikomori phenomenon is probably a scenario not achievable by one team of researchers, but the following study offers numerous insights and opportunities for those working on the phenomenon dealing with the problems that affect Hikikomori every day. The results obtained from this research are interesting because they are in line with those of other research conducted outside of Italy, and at the same time they are innova-

tive because from our analysis the emphasis is placed on the complicated relationship with professors, who often take on the role of the bully for Hikikomori, and the relationship with technology as a favorite tool for passing the time. We have focused the research on the investigation of the impact of the phenomenon on the environment and its sustainability of it in the community of southern Italy. The limitations of our research are mainly territorial. Further on, the research will be extended to the entire Italian territory, to have a not only regional but national framework to phenomenon. This will lead us to develop future research by overcoming methodological limitations by seeking direct forms of collaboration and interactions, not only this time with experts in the field and parents but also with people who have managed to overcome their condition of Hikikomori to have the most comprehensive view possible of the phenomenon studied. It will proceed for the continuation of the study to complete the Delphi survey, in which only the first round is presented on these pages. In addition, one will go on to integrate the research design with one or more QUAN-like methods. For this, it is intended to continue using an MM-type methodology, to be able to study the Hikikomori phenomenon and at the same time test this methodology, which is still underutilized in Italy. For future research, we would like to focus mainly on one of the topics to be able to describe the Hikikomori syndrome that comes to develop in Italy.

**Author Contributions:** Conceptualization, V.E. and F.A.; methodology, V.E. and F.A.; validation, V.E., F.A. and F.R.L.; formal analysis, V.E and F.A.; investigation, V.E. and V.D.; data curation, V.E., F.R.L. and F.A.; writing—original draft preparation, V.E. and V.D. writing—review and editing, V.E. and V.D.; visualization, V.D.; supervision, F.A and F.R.L. All authors have read and agreed to the published version of the manuscript.

**Funding:** This research received no external funding.

**Institutional Review Board Statement:** Not applicable.

**Informed Consent Statement:** Informed consent was obtained from all subjects involved in the study.

**Data Availability Statement:** The data presented are not publicly accessible. We gained access through joint work with the Hikikomori Italia Association.

**Conflicts of Interest:** The authors declare no conflict of interest.

**Ethics Statement:** The research was conducted with full respect for the participants and research partners. The data was anonymized so that it could not be linked to the respondents of the focus group and the Delphi survey. The results, before being released, were shared with the respondents so that they could see.

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
