# Peer review of "The Sustainability of Emerging Social Vulnerabilities: The Hikikomori Phenomenon in Southern Italy"

_sustainability, doi:10.3390/su15043869_

Round 1
Reviewer 1 Report
1. The abstract lacks the information about the research goal, methodology, study participants and the country/region where the analyses were conducted.
2. Introduction is clear, consistent and well-written
3. There is no information about the size of the focus group (of the parents included in the study)
4. Results and discussion are presented in an orderly and comprehensive way
5. The subject matter discussed in the article and the obtained results are interesting and important, also from the perspective of providing support for the individuals experiencing hikikomori and their families. Taking into account the cultural context of the analysed phenomenon is cognitively valuable as well
Reviewer 2 Report
I have read the article with great pleasure. I believe that the article is of great relevance in the extant literature of hikikomori, especially in Western European regions. The findings can provide evidence to future studies that are interested in aiming for potential interventions. However, there are several major points that need to be addressed before the text can be considered as publication.
Please refer to the pdf attached to read my comments.
